# Gluten Degradation by the Gut Microbiota of Ulcerative Colitis Patients

**DOI:** 10.3390/microorganisms11010012

**Published:** 2022-12-21

**Authors:** Emma Olivia Schultz Harringer, Juliana Durack, Yvette Piceno, Vibeke Andersen, Susan V. Lynch

**Affiliations:** 1Department of Medicine, Division of Gastroenterology, University of California San Francisco, San Francisco, CA 94143, USA; 2Molecular Diagnostics and Clinical Research Unit, IRS-Centre Soenderjylland, University Hospital of Southern Denmark, 6200 Aabenraa, Denmark; 3Institute of Regional Research, University of Southern Denmark, 5000 Odense, Denmark; 4Institute of Molecular Medicine, University of Southern Denmark, 5000 Odense, Denmark

**Keywords:** ulcerative colitis, gluten sensitivity, gluten-degrading microbes, *Bacillus*, *Enterococcus*

## Abstract

Several studies have reported improved disease symptomatology in ulcerative colitis (UC) patients consuming a gluten free diet. This observation coupled with diversity depletion in the gut microbiota of UC patients led us to hypothesize that UC-associated enteric microbes differentially metabolize dietary gluten to produce immunogenic products that promote inflammation. Gluten concentration in stool was determined using gluten-specific ELISA, and gluten intake was assessed by food frequency questionnaire (FFQ) in UC (*n* = 12) and healthy controls (HC; *n* = 13). Gluten-metabolizing bacteria were isolated on minimal media supplemented with 1% gluten from UC and HC and identified by 16S rRNA profiling. Cell-free culture media from gluten metabolizing gut bacterial isolates was assessed for immunogenicity in vitro using HT29 colonocytes. Compared to HC, UC patients did not consume gluten differently (Mann–Whitney; *p* > 0.10) and exhibited equivalent levels of gluten in their feces (Mann–Whitney; *p* = 0.163). The profile of gluten-degrading bacteria isolated from UC stool was distinct (Chi-square; *p* ≤ 0.0001). Compared with *Enterococcus* isolates, products of gluten degradation by *Bacillus* strains induced higher IL8 and lower occludin (Mann–Whitney; *p* = 0.002 and *p* = 0.059, respectively) gene expression in colonocytes irrespective of whether they originated from UC or healthy gut. Members of HC and UC microbiota exhibit gluten-degrading ability, metabolites of which influence genes involved in inflammation and barrier function in enteric colonocyte cultures. Preliminary findings of this study warrant further investigations into the mechanisms by which gut microbiota contribute to UC pathogenesis through gluten degradation.

## 1. Introduction

Ulcerative colitis (UC) is the most common form of inflammatory bowel disease (IBD) particularly in westernized nations, where diets and lifestyle have dramatically changed over the last several decades towards, amongst other transitions, a high intake of refined grains containing gluten [1,2,3,4].

UC is a debilitating, idiopathic gastrointestinal inflammatory disease that can lead to life-threatening complications including a higher risk for cancer [5]. Current treatment options include anti-inflammatories, immunomodulating biologics and/or surgery. Though pharmaceutical interventions may manage disease symptoms, they are not universal in their efficacy and there is a major need to develop a better understanding of factors that lead to maintenance of chronic inflammation in this patient population so that novel therapies may be developed.

A consistent feature of UC patients is microbiome perturbation [6]. Patients with UC characteristically exhibit compositionally distinct fecal and intestinal mucosal microbiomes depleted of microbial diversity commonly observed in age-matched healthy subjects [7,8]. Loss of gut microbial diversity results in loss of functional gene pathways encoded by these microbes, particularly metabolic pathways, resulting in altered metabolism of dietary substrates [9]. In addition to inflammation, increased intestinal permeability is found in 40–50% of patients with IBD, with the highest values for those with active disease [10]. Increased permeability can be assessed using occludin [11] and the overexpression of this tight junction protein considered a mucosal response to the increased permeability [12]. The inflammation in the gut of IBD patients can be assessed using IL-8 as a biomarker [13].

Gluten, a dominant component of Western diets [14], is more commonly associated with celiac disease and almost all knowledge of how gluten influences the immune system stems from these patients. In celiac disease patients, both undegraded gluten and gluten degraded to specific peptides, i.e., gliadin and glutenin, promote inflammatory responses in the gastrointestinal tract [15,16]. Other research has shown that gluten is largely resistant to cleavage by human gastrointestinal digestive enzymes [17], and that it is primarily intestinal microbes that metabolize gluten in the human gut [18].

Support to examine the relationship between UC and gluten comes from recent observations in patients with IBD, rheumatoid arthritis and psoriasis, who report reduced disease symptomology whilst consuming a gluten free diet (GFD) [19,20]. A recent study reported improvement in gastrointestinal (GI) symptoms such as abdominal pain, diarrhea and headaches in 70 UC patients on GFD [21]. Additionally, a study examining 1647 adults found that 65.6% of patients with IBD who implemented a GFD experienced an improvement of at least one clinical symptom, 38.3% reported fewer and less severe flares while abstaining from gluten and 23.6% required less medication to control disease [22].

Several studies have been conducted to investigate a possible connection between celiac disease and IBD with various results. Two studies did not find a higher prevalence of human leukocyte antigen (HLA) DQ2/8 [23,24] in patients with UC compared to healthy subjects, the necessary antecedent for developing celiac disease [25]. Elevated concentrations of anti-tissue-transglutaminase, anti-tTG antibodies, characteristic of celiac disease [15], have also been reported in IBD patients [26], suggesting that similar disease pathology may exist in UC patients.

These observations have led us to hypothesize that due to gut microbial species depletion, patients with UC are enriched for species that differentially metabolize dietary gluten to produce immunogenic products that promote inflammation. Our aim is to isolate gluten-degrading bacteria from the gut of UC patients and healthy subjects, transfer the degraded gluten products to a colonic cell-line and test the cells expression of occludin, to assess barrier function, and IL8 to assess inflammatory response.

## 2. Materials and Methods

### 2.1. Study Population and Sample Collection

Stool samples were collected from age-matched healthy participants (*n* = 13) and physician-diagnosed ulcerative colitis patients (*n* = 32) as previously described [7]. The University of California San Francisco Committee on Human Research approved the protocol for collection of stool samples used in this study. Subjects provided written informed consent. In this study, the participants were grouped based on disease severity and associated fecal bacterial community composition, previously classified [7]. Samples from the group with highest disease activity (UC *n* = 12, HC *n* = 13) (see Appendix A for patient characteristics) were used for gluten and microbiological assays performed in this study. Home stool samples were collected and maintained on ice until stored at −80 °C.

### 2.2. Assessment of Gluten Intake in Study Participants:

All subjects completed a Block 2005 Food Frequency Questionnaire (FFQ) which was used to assess gluten intake as combined consumption of white bread, bagels, English muffins and biscuits.

### 2.3. Quantification of Fecal Gluten Content

Amount of gluten in fecal samples was determined using the gluten ELISA (Allertek, Gainesville, FL, USA) kit. Protocol was followed per manufacturer’s instructions, except for dilution with Sample Dilution Buffer, which was reduced from 10× to 2×, due to anticipated lower concentrations of gluten in feces compared to food samples. Briefly, frozen stool (0.25 g) was weighed and resuspended in Sample Dilution buffer at 1:1 *w*/*v* ratio. Samples, standards and controls were loaded into a 96-well plate containing the 401.21 antibody, which recognizes both the gliadin and the glutenin fractions of gluten. Gluten concentrations were calculated against a standard curve based on 450 nm absorbance.

### 2.4. Isolation of Gluten Degrading Bacteria

Pooled healthy or UC fecal samples were cultured on M9 Minimal Salts X5 (M9; Sigma-Aldrich, Saint Louis, MO, USA) agar plates with added trace elements, MgSO_4_, CaCl_2_ and 1% gluten (Sigma-Aldrich, Saint Louis, MO, USA) under aerobic (AE), microaerophilic (MA) or anaerobic environments (AN) for 24 h. Anaerobic conditions were maintained in an anaerobic chamber (Coy Laboratory Products, Grass Lake, MI, USA). Microaerophilic atmosphere was generated using CampyGen and CO_2_Gen gas generating kits (BD, Franklin Lakes, NJ, USA). Putative gluten degrading isolates were validated by sub-culture onto M9 media with or without addition of filter-sterilized (0.2 μm) (Thermo Fisher, Waltham, MA, USA) 1% gluten for 72-hour under the oxygen availability conditions used for initial isolation.

### 2.5. Identification of Gluten-Degrading Microorganisms

Confirmed gluten metabolizing isolates were plated on Luria-Bertani (LB) agar (Sigma-Aldrich, Saint Louis, MO, USA) and incubated aerobically for 48 h at 37 °C. Species identification was determined by Sanger sequencing (QuintaraBio, San Francisco, CA, USA) of the full-length 16S rRNA gene amplified using 27F/1492R primers, and by using gDNA purified followed by QIAquick PCR Purification Kit (Qiagen, Hilden, Germany) from each isolate. The resulting sequences were classified using the ribosomal rRNA SILVA (https://www.arb-silva.de/, accessed on 1 May 2019) database [27]. Isolates were cryopreserved at −80 °C in LB broth containing 50% (*v*/*v*) sterile glycerol.

### 2.6. Epithelial Cell Assessment of Bacterial Products of Gluten Degradation

Gluten degrading bacteria isolated from UC patient and healthy subject feces were cultured overnight and diluted to an OD600 = 0.1 in M9 media + 1% gluten, prior to growth for 8 h under aerobic conditions. Cellular material was removed by centrifugation (13,000 rpm for 1 min) and supernatants were passed through a 0.22 μm syringe filter (Millipore, Burlington, VT, USA) to generate cell-free supernatants. Human epithelial colorectal adenocarcinoma HT29 cells were seeded at a confluent density in cell culture dishes with added McCoy’s 5A media (Thermo Fisher, Waltham, MA, USA) supplemented with 10% heat-inactivated FCS (USA Scientific, Ocala, FL, USA), 100 U/mL penicillin-streptomycin (Life Technologies, Carlsbad, CA, USA) 24 h before exposure to cell-free bacterial supernatants (SBS; 1:4 dilution) for 24 h. Total RNA from HT29 cells was isolated with RNAqueous^®^-Micro Kit (Thermo Fisher, Waltham, MA, USA) following manufactures instructions and contaminant DNA digested with DNase I (Sigma-Aldrich, Saint Louis, MO, USA) for 20 min at 37 °C. First-strand cDNA was synthesized using 200 ng RNA with High-Capacity RNA-to-cDNA TM Kit (Thermo Fisher, Waltham, MA, USA) according to the manufacturer’s instructions.

Real-time quantitative PCR was performed in triplicate with SYBR Green master mix (Life Technologies, Carlsbad, USA) in the QuantStudio™ 6 Flex Real-Time PCR System (Applied Biosystems, Waltham, MA, USA) using the primers for occludin (GATGAGCAGCCCCCCAAT [forward], GGTGAAGGCACGTCCTGTGT [reverse]) to assess barrier function and IL-8 (CTGAGAGTGATTGAGAGTGGAC [forward], AACCCTCTGCACCCAGTTTTC [reverse]) to assess inflammatory response. PCR reaction conditions were as follows: 50 °C for 2 min, 95 °C for 10 min (1 cycle); 95 °C for 15 s and 60 °C for 1 min (40 cycles); and a final melting curve cycle of 95 °C for 15 s, 60 °C for 1 min and 95 °C for 15 s. Gene expression was normalized to β-actin expression and relative fold change in expression determined by the 2-∆∆CT (where CT is threshold cycle) method using PBS exposed cells as control. β-actin primers were AAGATGACCCAGATCATGTTTGAGACC [forward] and AGCCAGTCCAGACGCAGGAT [reverse].

### 2.7. Statistical Analysis

Mann–Whitney test was used for group comparisons. GraphPad Prism version 6 (GraphPad Software. La Jolla California, CA, USA, www.graphpad.com, accessed on 1 February 2019) software was used for statistical analyses. Differences in gluten-degrading bacterial profiles were compared using Chi-square test.

## 3. Results

### 3.1. The Gut of UC Patients Retains Gluten Degrading Capability

Using data captured in the FFQ, gluten intake was assessed based on subject reported consumption of gluten containing foods and compared between healthy subjects and UC patients. Gluten consumption was not found to significantly differ between groups (Appendix A). A comparative analysis revealed that the concentration of gliadin and glutenin (the metabolites of gluten) in feces of subjects, as assessed by gluten-specific ELISA, did not differ between the two groups (Mann–Whitney; *p* > 0.05) (Figure 1 and Appendix A), suggesting that UC patient retain the ability to degrade gluten despite loss of microbial function in their gut microbiome.

### 3.2. Distinct Profiles of Gluten Degrading Bacteria Are Evident in UC and Healthy Feces

Bacteria capable of growth on minimal media supplemented with gluten as the sole carbon source were evident in feces from both healthy subjects and UC patients. Sequencing of full length 16S rRNA was performed for 37 isolates to determine whether distinct bacterial species were capable of gluten degradation in UC patients compared with healthy subjects. Classification using the SILVA database indicated that the profile of bacterial capable of degrading gluten in the gut microbiome of UC patients was significantly distinct from that observed in healthy subjects (Chi-square; *p* ≤ 0.0001; Figure 2). Bacterial isolates from patients with UC predominantly belonged to the *Enterococcus*. In comparison a greater variety of bacterial species were isolated from the gut microbiome of healthy subjects, including species belonging to *Escherichia*, *Bacillus* and *Enterococcus*. These data suggest that the healthy gut microbiome possesses a greater breadth of species capable of gluten degradation.

### 3.3. Products of Gluten Degradation by Enterococcus and Bacillus Isolates Induce Distinct Immune Response in Colonocytes Irrespective of Their Source

We hypothesized that gluten-degrading bacterial isolates from UC patients induce barrier dysfunction and pro-inflammatory response compared to HC isolates. To assess this, the cell-free products of gluten degradation by various UC and HC fecal bacterial isolates was compared. In a proof of principal experiment, we used IL8 (a neutrophil chemoattractant [28]) and occludin (an epithelial tight junction protein [29]) expression as a proxy for immune activation and barrier dysfunction following exposure of HT29 colonocytes to culture supernatants from *Bacillus* or *Enterococcus* isolated from stool of UC or HC subjects.

Higher IL8 expression was evident following exposure of colonocytes to the cell-free products of *Bacillus* isolates grown on gluten containing M9 media when compared with *Enterococcus* isolates (Mann–Whitney; *p* = 0.002; Figure 3A). However, cell-free products of gluten degradation by *Enterococcus* isolates trended towards increased occludin expression (Mann–Whitney; *p* = 0.059; Figure 3B). This finding was consistent irrespective of the source of gluten metabolizing strains (HC or UC; Appendix A) and indicated that these genera may contribute to inflammation if actively engaged in gluten metabolism in the gut. Contrary to our hypothesis we did not observe that UC isolates capable of growth on gluten as the sole carbon source were more inflammatory or capable of greater barrier disruption compared with their HC phylogenetic counterparts (Mann–Whitney; *p* > 0.05; Appendix A).

## 4. Discussion

Our observations suggest that UC patients do not differ in their consumption of gluten and retain the capacity to degrade ingested gluten. However, we note that the diversity of bacteria capable of gluten degradation is diminished in UC patients and consists primarily of *Enterococcus* and *Bacillus* species. This suggests that the products of these species’ metabolism of gluten predominate the luminal contents of UC patients.

This is a first report, to our knowledge, linking the gut microbiome of UC patients to gluten degradation and the possible mechanism of gluten sensitivity. Depletion of bacterial diversity is a well-documented signature of the UC gut microbiome [6]; this is reflected in the reduced diversity of gluten degrading bacteria isolated from UC subjects. Gluten degradation by enteric microbiota has been observed in celiac disease [30], and the byproducts of bacterial metabolism of gluten are known to contribute to the immune activation [19].

*Bacillus* species have been shown to hydrolyze gluten well [31], and inactivated *Bacillus* species have been shown to improve the symptoms of UC [32] and alter the production of both immune activating and anti-inflammatory cytokines and chemokines [33]. Our observations in this study suggest a higher frequency of gluten-degrading *Bacillus* in healthy subjects compared to UC patients. In vitro isolates from both subject groups exhibit comparable capacity to induce IL8 and occludin expression in cultured HT29 colonocytes. On the contrary, members of genus *Enterococcus* have been shown to contribute to development of intestinal inflammation in mice [34,35] and have been found increased in feces of UC patients [36]. Our findings indicate a higher frequency of gluten-degrading *Enterococcus* in the stool of UC patients. However, in vitro findings indicated no difference in the capacity of gluten metabolism products derived from a UC-associated *Enterococcus* strain to induce either IL8 or occluding when compared with an isolate from a healthy subject. Increased occludin expression was observed following exposure to the gluten metabolism products of the *Enterococcus* isolates, compared to those from *Bacillus*, though whether this reflects increased barrier disruption is unclear. Overall, our cell-culture based screen for pro-inflammatory potential of gluten metabolism by enteric bacteria does not exclude the possibility that UC isolates may elicit distinct epithelial responses compared to isolates from similar genera derived from healthy subjects in an already inflamed gut of UC patients or in association with other UC microbiota community members. It does indicate that the products of gluten degradation by distinct enteric bacteria are capable of inducing expression of genes associated with inflammatory immune activation and barrier dysfunction that may contribute to inflammation in UC patients.

Our observations are encouraging and, although very preliminary, these warrant further investigation into the mechanisms by which diet interacts with the gut microbiome to influence intestinal inflammation in UC patients. Limitations of this study include sample size, culture-based methods, and the use of an in vitro cell line assay. The findings of this study need to be further evaluated in a larger cohort of subjects. Additionally, investigation using community-based approaches such as functional metagenomics and metatranscriptomics will need to be incorporated into future studies for more “in depth” evaluation of gluten degrading members of the gut microbiome and how they relate to UC pathogenesis. Even though the results are preliminary, they show that the gut microbiome of IBD patients respond differently to gluten in their diet, potentially contributing to inflammation associated with the disease. Patients with IBD are very motivated to diet interventions as this gives them influence in their otherwise very disease-controlled life. Preliminary findings of this study warrant further investigations into the mechanisms by which gut microbiota contribute to UC pathogenesis through gluten degradation to provide patients with better understanding of ways to manage their disease.

## Figures and Tables

**Figure 1 microorganisms-11-00012-f001:**
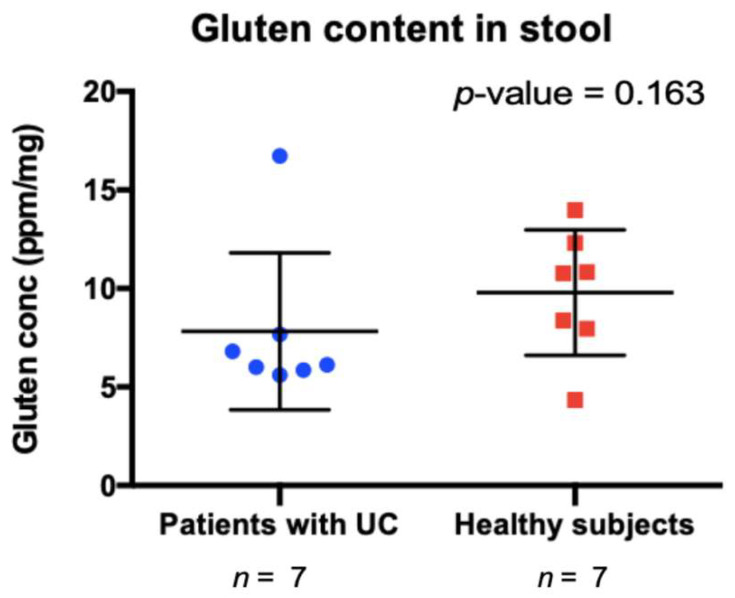
Undegraded gluten content in stool samples from patients with ulcerative colitis (UC) and healthy controls (HC). *p*-value obtained using Mann–Whitney test.

**Figure 2 microorganisms-11-00012-f002:**
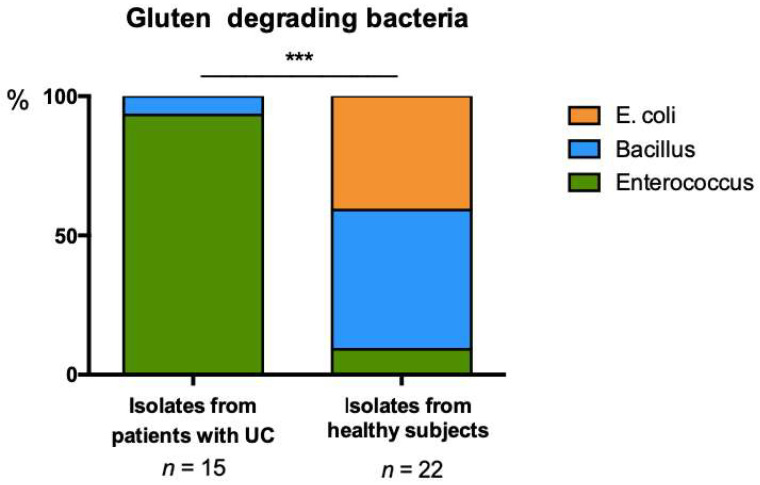
Distribution of gluten degrading bacteria isolated from stool of UC and HC subjects on minimal media supplemented with 1% gluten, Chi-square; *** *p* ≤ 0.0001.

**Figure 3 microorganisms-11-00012-f003:**
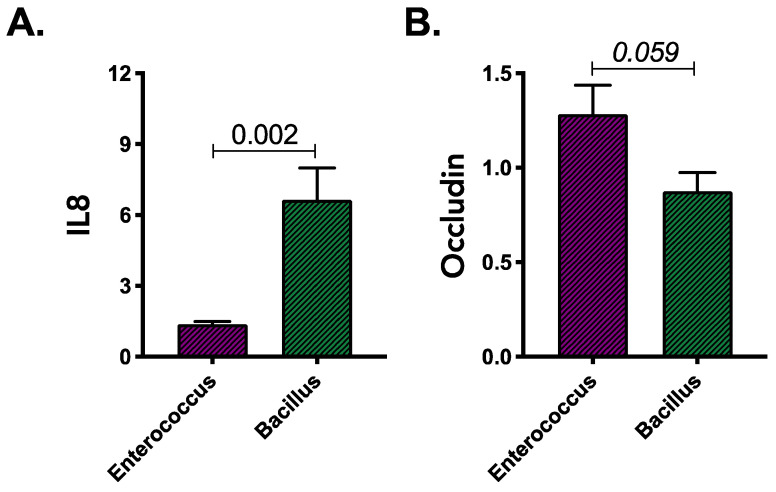
Expression of (**A**) IL8 and (**B**) occludin genes in colonic epithelial cells exposed to gluten degradation products produced by *Bacillus* and *Enterococcus* isolates. Results were obtained from two independent experiments and based on combined observations from 2 strains for each genera. Statistical significance was determined using Mann–Whitney test.

## Data Availability

The data underlying this article will be shared on reasonable request to the corresponding author.

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
