# Peer review of "Gluten Degradation by the Gut Microbiota of Ulcerative Colitis Patients"

_microorganisms, 2022, doi:10.3390/microorganisms11010012_

Round 1

Reviewer 1 Report

The authors described gluten degradation by the gut microbiota of ulcerative colitis patients. The experiments appear to be appropriate and some results are enlightening. However, the results and conclusion need to be improved. I recommend major revision of the paper before publication.

1. Line 20 The p value should be italic. ‘p > 0.10’--->‘p > 0.10’. Please revise it in the whole manuscript.

2. Line 23 ‘Enterococcus--->‘Enterococcus’, ‘Bacillus’--->‘Bacillus’. Plese make the species or genus names italic in the whole manuscript.

3. Line 72 ‘...depletion, patients with’

4. Line 74 More introduction about the significant roles of IL8 and occludin in inflammation should be added if authors mainly focused on these proteins to corroborate their hypothesis.

5. Lines 71-74 In addition to mentioning the hypothese being tested, for better comprehension, briefly mentioning the main aim or conclusions is recommended.

6. Lines 112. Please rewrite for better understanding. Currently, too much ‘using’ appeared.

7. Line 144 ‘...was used for between group compositions.’

8. Line 146 ‘ Differences in gluten-degrading bacterial profiles were compared using Chi-square test.’

9. Line 171 Again, names of species should be italicized.

10. Line 191 ‘Whereas....’ --->‘However,

11. Line 193 ‘Mann-Whitney; p=0.059 ; Fig 3B?’

12. Lines 238-240 ‘It does indicate that the products of gluten degradation by distinct enteric bacteria are capable of inducing expression of genes associated with inflammatory immune activation and barrier dysfunction that may contribute to inflammation in UC patients.’ The authors only carried out experiments checking for the gene expressions of IL 8 and occludin. Have authors tried to check other inflammation response-related gene expressions or inflammatory mediators to support this deduction?

Author Response

Dear reviewer 1

Thank you so much for your comments!

Comments 1-3 + 6-11: please see the manuscript for grammatical edits. 

Comment 4+5: I've added a section on Il-8 + occludin and our aims to the introduction. 

Comment 12: Unfortunately not. Due to time restrictions we only tested the two, but as you're indicating it would be relevant to test more to gain a more solid foundation for the deduction. 

Again, thank you for your time and the comments!

My best, 

Emma Harringer, MD

Reviewer 2 Report

See document attached

Author Response

Dear reviewer 2

Thank you so much for the comments!

I've attached your pdf file with my comments, please see the manuscript for the edits.

Again thank you for your time and comments, 

My best

Emma Harringer, MD

Round 2

Reviewer 1 Report

The authors have revised the manuscript according to the comments.

Reviewer 2 Report

see document attached
